# Utility Analysis about Log Data Anomaly Detection Based on Federated Learning

**Tae-Ho Shin** and **Soo-Hyung Kim** *

Interdisciplinary Program of Information Security, Chonnam National University,
Gwangju 61186, Republic of Korea; edensth@naver.com
* Correspondence: shkim@jnu.ac.kr

**Abstract:** Logs that record system information are managed in anomaly detection, and more efficient anomaly detection methods have been proposed due to their increase in complexity and scale. Accordingly, deep learning models that automatically detect system anomalies through log data learning have been proposed. However, in existing log anomaly detection models, user logs are collected from the central server system, exposing the data collection process to the risk of leaking sensitive information. A distributed learning method, federated learning, is a trend proposed for artificial intelligence learning regarding sensitive information because it guarantees the anonymity of the collected user data and collects only weights learned from each local server in the central server. In this paper, we executed an experiment regarding system log anomaly detection using federated learning. The results demonstrate the feasibility of applying federated learning in deep-learning-based system-log anomaly detection compared to the existing centralized learning method. Moreover, we present an efficient deep-learning model based on federated learning for system log anomaly detection.

**Keywords:** federated learning; deep learning; log analysis; anomaly detection





## 1. Introduction

Logs that record system information are currently generated at a scale of about 50 GB per hour in distributed data processing systems, such as the Alibaba Cloud, making it very difficult to manually identify major logs for anomaly detection, even with search utilities [1]. Even if the anomaly detection task is performed using automatic detectors, administrators, such as security personnel, must investigate many security warnings unnecessarily because not all logs are threats. According to Cisco's 2019 survey, 41% of the 3540 organizations surveyed received more than 10,000 security alert notifications daily, and only 50.7% of the total alerts were investigated due to the limited capabilities of security managers. Significant threat alerts comprised only 24.1% of the total alerts [2]. Thus, anomaly detection has improved over the years by improving individual detection tools for risk logs or adjusting alarm risk ranking classification, but detection for anomaly logs is still insufficient [3].

Deep learning-based log anomaly detection models that automatically detect system anomalies through log data have recently been proposed to reduce such problems [4]. However, existing methods can cause data leakage when collecting logs recorded in user systems in the central server for deep learning [5]. In particular, there are cases in which logs with sensitive information, such as user accounts, are recorded without proper masking or filtering by individuals and companies, causing irreversible damage if the log data are leaked in the data collection process [6].

Therefore, this paper analyzes the applicability of federated learning to deep learning-based log data anomaly detection. Unlike existing artificial intelligence models, federated learning learns global models by transferring only the weights trained on each local server

to the central server without directly collecting data [7]. Therefore, it is possible to prevent leakage of log data containing sensitive information, and because the raw log data required for deep learning are not concentrated in one place, the leakage damage is relatively low. Furthermore, we present a deep-learning model that achieves better performance in log anomaly detection.

The main contributions are as follows:

(1)   We analyze the performance of log anomaly detection through existing general deep learning processes.
(2)   We apply a federated learning framework to the models in (1) to compare their performance with existing learning processes and analyze the applicability of federated learning for log anomaly detection.
(3)   Finally, we demonstrate that the hybrid model combining the two models perform better than the application of a single deep learning algorithm, CNN1D, LSTM, in log anomaly detection.

The remainder of this paper is organized as follows. Section 2 describes studies related to federated learning and deep learning-based log anomaly detection. Section 3 details the proposed federated learning-based log anomaly detection model, and Section 4 analyzes and evaluates the experiments on that model. Finally, Section 5 presents the conclusions and future studies.

## 2. Related Works

### 2.1. Federated Learning

Unlike existing artificial intelligence learning, federated learning does not directly collect user data in the central server; instead, learning proceeds in each user's environment. Then, the weights derived through learning in each user environment are transmitted to the central server, and the global model updates and aggregates the transmitted weights. Thus, data anonymity is guaranteed because the weights are transmitted rather than the data [8].

Regarding the global model update method, two methods can be applied, federated stochastic gradient descent (FedSGD) and federated averaging (FedAVG). First, FedSGD collects parameters learned from one update communication from a central server, then calculates the average parameter values for all clients to update the global parameters, sending the results back to all local servers. Afterward, the corresponding process is repeated until the convergence condition of the corresponding parameter set is satisfied [9]. Although these methods are simple and computationally efficient, one local update must be reflected every time in a single learning process, and FedAVG has been proposed for this purpose. Unlike FedSGD, FedAVG can reduce global update time by delivering parameter values to servers after each local server repeatedly learns a certain number of rounds. Therefore, the experiments on federated learning were conducted by applying this method [10].

Federated learning can be classified into horizontal federated learning, vertical federated learning, and federated transfer learning, in addition to the commonality that data is generated in the user environment and that it is coordinated and managed overall on a central server. Horizontal federated learning is when the same feature is used for more than one user dataset, and vertical federated learning is when the dataset is the same but the features used are different. Federated transfer learning is a technique that improves performance through similar learning methods due to a lack of commonality in both data and features [9]. The method applied to the model proposed in this paper is horizontal federated learning, as it extracts the same features for different log datasets from two or more local servers.

### 2.2. Deep Learning-Based Log Anomaly Detection

Deep learning-based anomaly detection models have been proposed to solve the problem of conventional log anomaly detection [4]. The general framework to apply the existing anomaly detection model is illustrated in Figure 1.

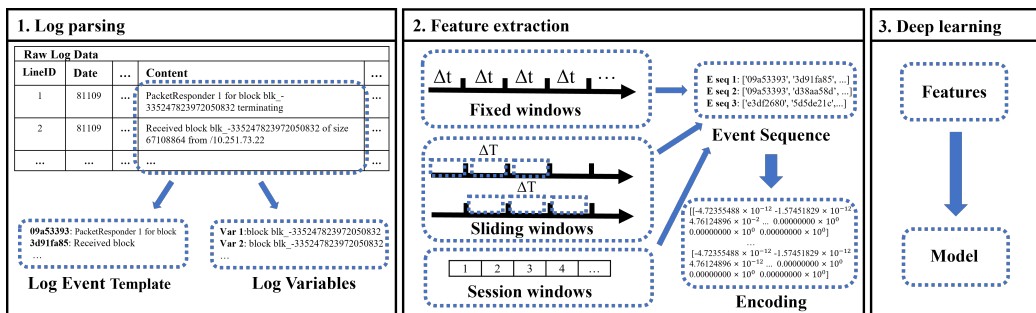

**Figure 1.** Deep learning-based log anomaly detection framework.

### 2.2.1. Log Parsing

Logs are typically text in the form of messages with various fields, including the recording time and system processing content. Therefore, extracting specific values by filtering unnecessary information for feature extraction from unstructured raw log data is necessary.

Log parsing extracts system messages from several fields in each log with time stamps and major system processing messages recorded, as depicted in Figure 1. Then, they are extracted separately from the system message into event-based log event templates and identifier-based log variables [11]. For example, in the *Receiving block blk*_335247823972050832 log, *Receiving* and *block* are extracted as a log event template, and *blk*_335247823972050832 is extracted as a log variable, as displayed in Figure 1. Each parsed item is clustered according to the following grouping method and is used as a feature for deep learning.

### 2.2.2. Pre-Processing and Feature Extraction

Parsed log sequences must be encoded into numerical features to be input into deep learning. Thus, the parsed logs are first grouped into sequences using the fixed window, sliding window, and session window grouping methods, as illustrated in Figure 1. The fixed window method groups them by the time stamp($\Delta t$) of each log. Sliding windows are grouped by time stamp($\Delta T$) and the transmission interval. In this case, "windows" means a fixed range, and the occurrence time range is set according to the length of the set windows. However, if the windows are too short, deep learning models have difficulty detecting anomalies that span multiple sequences, and if they are too long, the sequence may contain multiple anomalies, confusing the detection scheme [12]. Finally, unlike the other two methods, the session window method groups them according to identifiers rather than time order [5]. For example, the session window method groups event templates based on identifiers, a unique variable in each log.

Finally, the grouped log sequence is encoded using the term frequency-inverse document frequency (TF-IDF) method. The TF-IDF is primarily used to calculate the similarity of a search system or document and is a method of weighting the target word in that document through the frequency and importance of the target word [13].

The TF-IDF is obtained through the document($d$), word($t$), and total number of documents($n$). When the total frequency of target words in the document is $f(t, d)$, the equation for the word frequency $tf$ is presented in (1). In this case, $max\{f(w, d) : w \in d\}$ indicates the number of all words appearing in the document:

$$tf(t, d) = 0.5 + \frac{0.5 \times f(t, d)}{max\{f(w, d) : w \in d\}} \tag{1}$$

Then, $idf$, a value indicating how often one word appears in common across a set of documents, can be obtained by dividing the total number of documents by the number of documents that include the target word and converting it to a logarithm. In this case, $|D|$ denotes the number of all documents, and $|d \in D : t \in d|$ denotes the number of documents that include the target word. The formula is presented in (2):

$$idf(t,D) = log(\frac{|D|}{|d \in D : t \in d|})$$ (2)

Finally, $tfidf$ is expressed as (3) through (1) and (2):

$$tfidf(t,d,D) = tf(t,d) \times idf(t,D)$$ (3)

Log sequences encoded through the corresponding method are input into the deep learning model to perform learning for log anomaly detection.

### 2.2.3. Deep Learning For Log Anomaly Detection

The extracted features are input into the deep learning model to proceed with learning, and the normalities and abnormalities of the logs are classified based on the learned model. In this paper, by applying the following algorithms of models applied in the previous studies, we analyze the applicability of federated learning and compare the performance of each model.

(1) The Long Short Term Memory (LSTM) network is a series of recurrent neural network algorithms for time-series data processing and text recognition, and studies have applied them to log data with text. Du et al. proposed Deeplog, which models sequential patterns by applying the LSTM and identifies anomalies in the log [14]. Zhang et al. proposed LogRobust, which detects anomalous logs by applying the attention mechanism-based bidirectional LSTM [15]. Meng et al. proposed LogAnomaly, which identifies anomalous logs based on sequential and quantitative log information [16].

(2) The One-dimensional Convolutional Neural Network(CNN1D) algorithm can be applied to sequential data, unlike the two and three-dimensional CNN algorithms, because it calculates in one direction, and this method derives excellent performance in text classification through model design [17]. In addition, Lu et al. proposed a 1D CNN-based log anomaly detection model that uses fewer parameters than the models of the existing family of recurrent neural networks [18].

(3) Recently, a hybrid model that combines two or more deep learning models has been proposed, and various studies have been conducted, especially in text classification and prediction problems, such as those concerning log data [19–21]. Among them, in text classification research, a hybrid CNN model was effective in extracting local features, such as words, and an LSTM model was effective in extracting features for contextual information or word order to derive higher performance than the existing single models [19]. Therefore, we perform anomaly detection on log data by combining the LSTM model with the proposed CNN1D model and comparing the performance results with those of existing models.

## 3. Proposed Federated Learning Model

This paper proposes an anomaly detection model for federated learning-based log data. Figure 2 presents the framework of the model proposed in this paper.

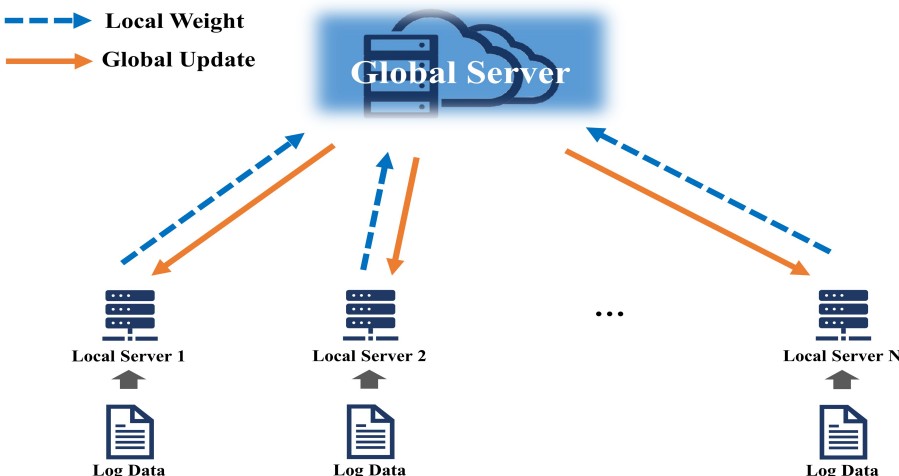

**Figure 2.** Framework of federated learning process.

### 3.1. Federated Learning Framework

As illustrated in Figure 2, the framework has a local server that learns each local dataset and a global server that aggregates the weights of the models sent from the local server, and the same deep learning model is applied to both servers. Each local server learns according to epochs in which features extracted from local data are set. Then, the corresponding learning process is repeated per the number of repetitions set according to the FedAVG method, and the average for the total weight is calculated. Next, the average weights generated by the learning are sent to the global server, and the original data from the local server are not directly transmitted to the global server. The global server averages and updates the aggregated weights at all local sites. The updated weights are sent back to each local model. Afterward, the process is repeated according to the set round, and model learning at the final global server is completed.

### 3.2. Anomaly Detection Based on CNN1D

Figure 3 displays the architecture of the CNN1D model to be applied to the proposed federated learning. In this study, we trained a single CNN1D model with the following architecture: an embedding layer with dimensions of $50 \times 16$, followed by a single CNN1D layer with a filter size of $5 \times 128$, and a dense output layer with two units for normal or anomaly prediction tasks. Each input-encoded log depends on the dataset. First, an encoded learning matrix is input into the embedding layer. Log sequences grouped in identifier order by applying the session window method are encoded using the TF-IDF method. For example, log sequences grouped into $[09a53393, 3d91fa85, 09a53393, \ldots]$ are encoded in a vector set of $[-4.723555488 \times 10^{-12}, -1.57451829 \times 10^{-12}, \ldots]$ and then entered in the embedding layer. After calculating the features embedded in a single convolutional layer of size $5 \times 128$, the rectified linear unit activation function is applied to prevent gradient loss for nonlinear conversion, and the reduced matrix is input into the fully connected (FC) layer by removing the low-correlation feature in the max-pooling layer. In addition, a 0.3% dropout layer was added to prevent overfitting before inputting it into the output layer. Finally, after mapping the matrix input through the FC layer in one dimension, the softmax layer is applied to output the classification result to classify the anomaly or normal data based on the learned weight. The equation is provided in (4), and the model parameter settings for each layer are listed in Table 1.

$$Softmax_i = \frac{e^{a_i}}{\sum\limits_{k=1}^{T} e^{a_k}} \tag{4}$$

where $a_i$ represents the log sequence input to the softmax layer, and $T$ is the total length of the log sequence.

**Table 1.** Specific parameters of the CNN1D model used in our study.

| Layer | Output |
|---|---|
| Input: encoded log | - |
| Embedding | $50 \times 16$ |
| CNN1D layer: 128 | $10{,}368 \times 128$ |
| Dense layer: 128 | $129 \times 128$ |
| Max Pooling layer: 2 | 0 |
| Dropout layer(0.3) | 0 |
| FC: 2 | 258 |
| Softmax layer | |

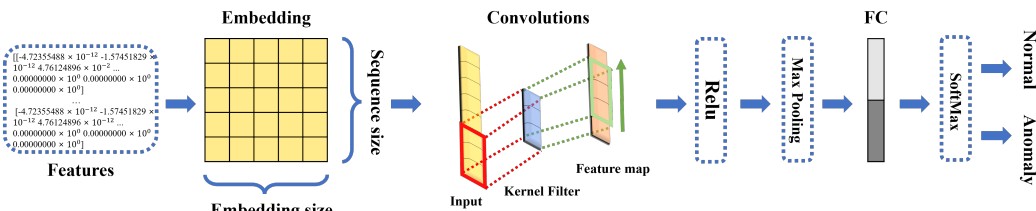

**Figure 3.** CNN1D-based log anomaly detection model architecture.

### 3.3. Anomaly Detection Based on LSTM

Figure 4 displays the architecture of the LSTM model to be applied to the proposed federated learning. In this study, we trained a single LSTM model with the following architecture: an embedding layer with dimensions of $50 \times 16$, a single LSTM layer with 128 units, and a dense output layer with two units for normal or anomaly prediction tasks. Each input-encoded log depends on the dataset. Each LSTM cell has three gates: the input, forget, and output. First, preprocessed features are input into the embedding layer to be embedded and vectorized. Then, the values entered in the hidden layer comprise 128 hidden nodes circulated by each LSTM cell to calculate the weight of the information in the vectorized log sequence. Finally, after 1D mapping of the matrix input through the FC layer, all weights for the hidden layer are added, and the softmax layer is applied for classification output to perform classification on the anomaly or normal data. Table 2 lists the model parameter settings for each layer.

**Table 2.** Specific parameters of the LSTM model used in our study.

| Layer | Output |
|---|---|
| Input: encoded log | - |
| Embedding | $50 \times 16$ |
| LSTM layer: 128 | $117{,}760 \times 128$ |
| FC: 2 | 258 |
| Softmax layer | |

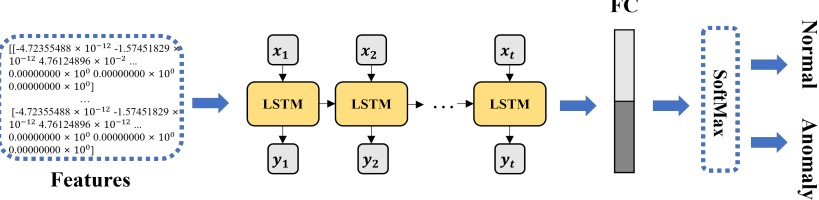

**Figure 4.** LSTM-based log anomaly detection model architecture.

*3.4. Anomaly Detection Based on CNN1D-LSTM*

Figure 5 depicts the architecture of the CNN1D-LSTM model to be applied to the proposed federated learning. In this study, we trained the 1D CNN-LSTM model, a hybrid combining two previously described single models using the same parameter values. Each input-encoded log depends on the dataset. First, the preprocessed features are computed through the embedding layer to the pooling layer to output a fixed-length feature set, like in the CNN1D model. Then, the corresponding feature sets are input to an LSTM model, consisting of 128 hidden layers, to calculate the weights for information in the vectorized log sequence. The features extracted through the two fused models are mapped one-dimensionally through the FC layer, and classification is performed on the anomaly or normal data through the softmax layer.

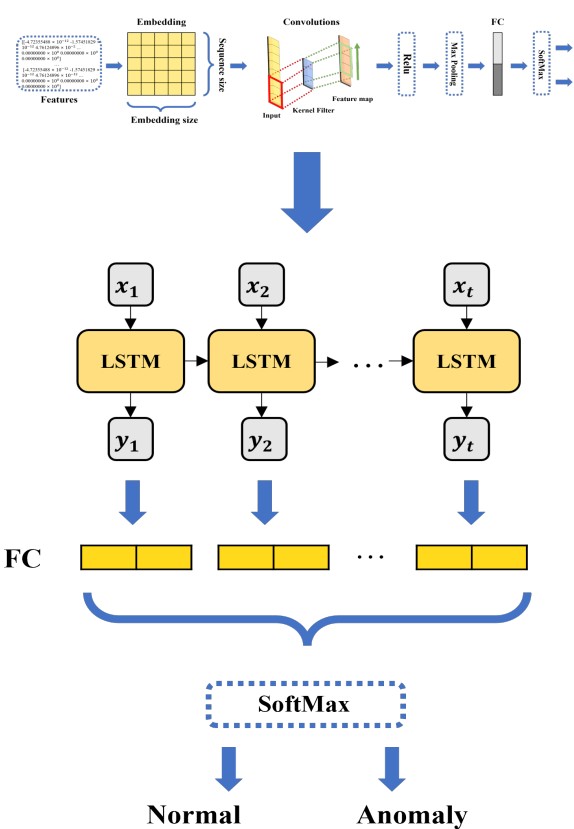

**Figure 5.** CNN1D-LSTM-based log anomaly detection model architecture.

## 4. Performance Evaluation

This section presents experiments to evaluate the performance of the proposed method. First, we describe the datasets and experimental settings. Then, we examine the applicability of federated learning using the performance comparisons between the centralized and federated learning methods through the described models. Finally, we compare the performance of each deep learning model.

*4.1. Dataset*

This paper employs two public datasets for the experiments, the Hadoop Distributed File System (HDFS) [22] and Blue Gene/L (BGL) [12] log datasets outlined in Table 3. The HDFS dataset consists of a total of 11,175,629 log messages generated over 200 days of experiments on Amazon EC2, of which an anomaly log is contained at a rate of about 2.9%. The dataset was recorded according to the identifier (*block_id*); thus, the experiment was conducted using the session window method. The BGL dataset consists of 4,747,963 log messages recorded by the Lawrence Livermore National Labs BGL supercomputer system,

with approximately 0.07% containing anomalous logs. In the case of the dataset, unlike HDFS, there is no identifier record, so the experiment was conducted through the sliding window method.

**Table 3.** Log datasets.

| Dataset | Train | Test | Total |
|---------|-------|------|-------|
| HDFS | 8,940,503 | 2,235,126 | 11,175,629 |
| BGL | 3,798,370 | 949,593 | 4,747,963 |

Each dataset was partitioned for learning on five local servers configured for federated learning, and the overview is presented in Tables 4 and 5.

**Table 4.** HDFS datasets for each local server learning.

| HDFS (Local) | Train | Test | Total |
|--------------|-------|------|-------|
| Client 1 | 1,788,100 | 447,026 | 2,235,126 |
| Client 2 | 1,788,100 | 447,026 | 2,235,126 |
| Client 3 | 1,788,100 | 447,026 | 2,235,126 |
| Client 4 | 1,788,100 | 447,026 | 2,235,126 |
| Client 5 | 1,788,100 | 447,025 | 2,235,125 |

**Table 5.** BGL datasets for each local server learning.

| BGL (Local) | Train | Test | Total |
|-------------|-------|------|-------|
| Client 1 | 759,674 | 189,919 | 949,593 |
| Client 2 | 759,674 | 189,919 | 949,593 |
| Client 3 | 759,674 | 189,919 | 949,593 |
| Client 4 | 759,674 | 189,918 | 949,592 |
| Client 5 | 759,674 | 189,918 | 949,592 |

*4.2. Environmental Setup and Evaluation*

The experiment was conducted using Windows 10 on an Intel (R) Core (TM) i7-8700 CPU, with an NVIDIA GeForce RTX 2080 Ti 11 GB GPU in a 16 GB RAM environment. Feature extraction and deep learning were conducted using Python 3.8.13 and the Tensorflow-gpu 2.9.1 module.

Federated learning is designed to use distributed data that cannot be collected directly from a central server, but in the research and development phase, it is convenient to design the model for download and manipulation on a local server, so we experimented in this way. The experiment was conducted on the premise that there was no network error between each client server. All three models applied to learning are classified as normal or abnormal, as in previous studies [14–16,18]; thus, the loss function was *categorical cross-entropy*, and *Adam* was employed as the optimizer. The learning rate was set to 0.01, and the batch size was set to 128.

The weighted F1-score [23] was used as an evaluation metric, a weighted average according to the number of data by class after calculating the F1-score through precision and recall, commonly used for classification models. These indicators derive values between 0 and 1, where values closer to 1 indicate higher performance (calculated via (4), (5), and (6)):

$$Precision = \frac{TP}{TP + FP} \tag{5}$$

$$Recall = \frac{TP}{TP + FN} \tag{6}$$

$$F1\text{-}score = \frac{2 \times Recall \times Precision}{Recall + Precision} \tag{7}$$

True positive (*TP*) represents the number of correctly predicted abnormal samples. False positive (*FP*) indicates the number of normally predicted abnormal samples, and false negative (*FN*) represents the number of normally predicted abnormal samples.

### 4.2.1. Federated Learning Performance Analysis

In the case of the proposed federated learning model, to derive the optimal performance and learning time for each model in each local server, the process of deriving the average value after learning 10 iterations according to the FedAVG method was performed for 10, 15, and 20 epochs and weight normalization. The process was conducted at thresholds in the range of 0.2 to 0.5. Values less than 0.2 and 0.6 or more were not normally weighted for the threshold, so the values were aggregated from the 0.2 threshold. The average and standard deviation of the derived values were aggregated to compare the evaluation indicators of each model. For the evaluation indicators, the values were compared by rounding to the fifth decimal place. The change in weighted F1-score mean values for the threshold and epoch of each model by dataset and the comparison results are provided in Figure 6.

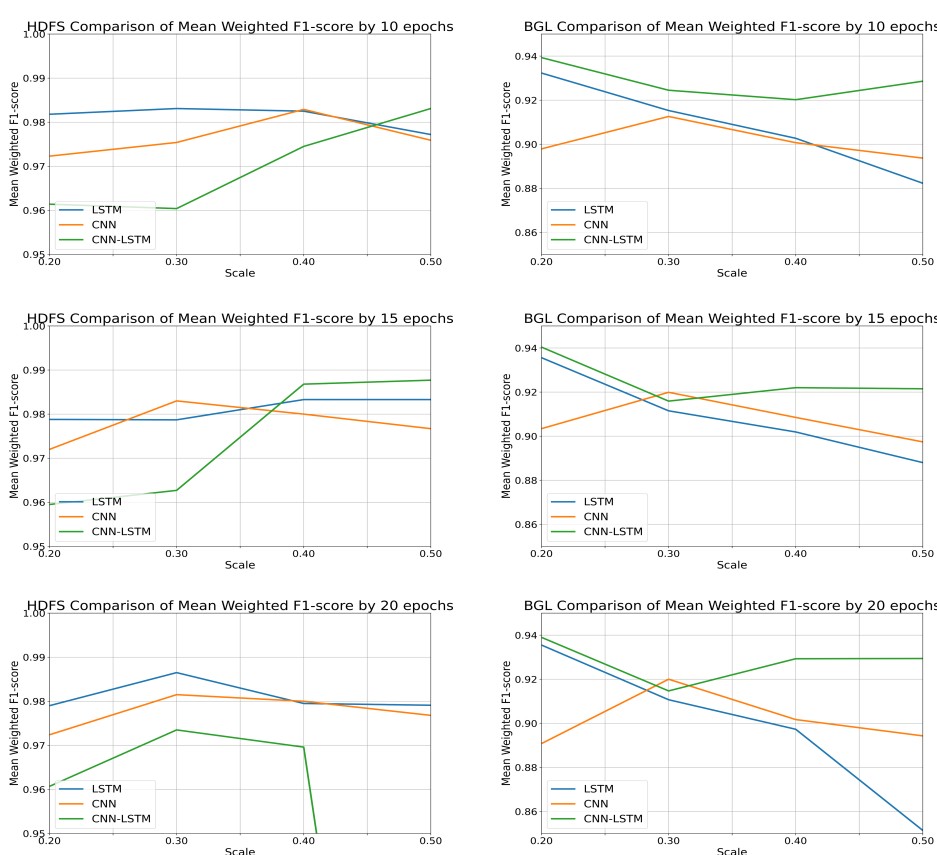

**Figure 6.** Weighted F1-Score comparison results by models.

According to these results, most models display a lower learning efficiency after the 0.4 threshold for both datasets. For the HDFS dataset, CNN1D models derive the highest average weighted F1-score of about 0.9830 at a 0.3 threshold and 15 epochs. The LSTM models derive about 0.9865 at a 0.4 threshold and 15 epochs, and the CNN1D-LSTM models derive about 0.9877 at a 0.5 threshold and 15 epochs. For the BGL dataset, CNN1D models derive the highest average weighted F1-score of about 0.9200 at a 0.3 threshold and 20 epochs. The LSTM models derive about 0.9865 at a 0.2 threshold and 15 epochs, and the CNN1D-LSTM models derive about 0.9404 at a 0.2 threshold and 15 epochs. These results confirm the possibility of log anomaly detection with federated learning through performance comparisons with the existing centralized learning methods.

### 4.2.2. Performance Comparison

Experiments were conducted in the same environment to compare performance with the existing centralized learning method. For the centralized learning method, after learning with epochs of 10, 15, and 20 for the two datasets, the average and standard deviation values of the weighted F1-score were calculated, and the results were rounded to the fourth decimal place.

For the HDFS dataset, the LSTM model derives approximately 0.990, 0.990, and 0.990 in 10, 15, and 20 epochs, respectively. The CNN1D model derives approximately 0.978, 0.990, and 0.990 in 10, 15, and 20 epochs, respectively, and the CNN1D-LSTM model derives 0.978, 0.990, and 0.990 in 10, 15, and 20 epochs, respectively.

For the BGL dataset, the LSTM model derives approximately 0.969, 0.971, and 0.970 in 10, 15, and 20 epochs, respectively. The CNN1D model derives approximately 0.946, 0.946, and 0.950 in 10, 15, and 20 epochs, respectively, and the CNN1D-LSTM model derives 0.970, 0.971, and 0.972 in 10, 15, and 20 epochs, respectively.

The comparison results between the average weighted F1-score value are listed in Tables 6 and 7. The comparison results by model are presented in Figure 7.

**Table 6.** Performance comparison by learning methods about the HDFS dataset.

| Model | Method | F1 Mean |
|---|---|---|
| CNN1D | Centralized | $0.987 \pm 0.005$ |
| | Federated | $0.983 \pm 0.008$ |
| LSTM | Centralized | $0.990 \pm 0.0003$ |
| | Federated | $0.986 \pm 0.004$ |
| CNN1D-LSTM | Centralized | $0.991 \pm 0.0005$ |
| | Federated | $0.988 \pm 0.006$ |

**Table 7.** Performance comparison by learning methods about the BGL dataset.

| Model | Method | F1 Mean |
|---|---|---|
| CNN1D | Centralized | $0.947 \pm 0.002$ |
| | Federated | $0.920 \pm 0.007$ |
| LSTM | Centralized | $0.970 \pm 0.002$ |
| | Federated | $0.937 \pm 0.002$ |
| CNN1D-LSTM | Centralized | $0.971 \pm 0.001$ |
| | Federated | $0.940 \pm 0.002$ |

The results reveal that the CNN1D-LSTM model yields the highest average weighted F1-score value of the two learning methods for system-log anomaly detection. In HDFS datasets, the proposed federated learning method for CNN1D, LSTM, and CNN1D-LSTM models exhibits 99.6%, 99.6%, and 99.7% performance compared to the existing centralized learning method, respectively. In BGL datasets, the proposed federated learning method for CNN1D, LSTM, and CNN1D-LSTM models displays 97.1%, 96.6%, and 96.8% performance compared to the existing centralized learning method, respectively.

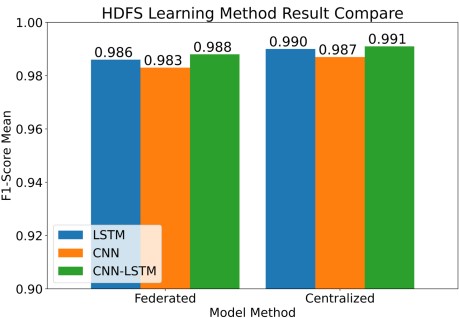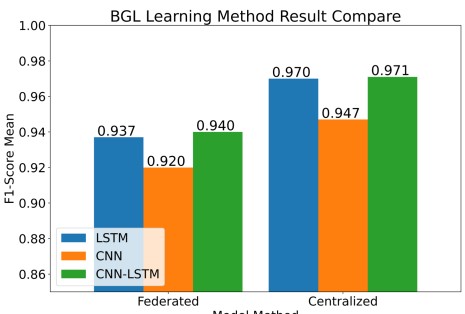

**Figure 7.** Performance comparison by deep learning models for datasets.

Although federated learning does not perform as well as general artificial intelligence learning because it aggregates the weights of local models trained separately from the distributed data, the results indicate a slight difference regarding all models used in the experiment when comparing performance with conventional learning methods. The corresponding results demonstrate that the model with federated learning is sufficiently competitive, considering the prevention of leakage of sensitive information during deep learning using log data. In particular, CNN1D-LSTM models derive the highest performance in both federated and centralized learning methods, deriving higher detection performance by extracting features more comprehensively than the other two models through advantages in local feature extraction and sequence data classification through LSTM. Furthermore, as found in the previous study [19], the application of the hybrid model rather than a single model performs better even for log data, which is similar to text data.

## 5. Conclusions

This paper analyzed the utility of log data anomaly detection based on federated learning. The results were compared with the existing centralized method under the same conditions for model verification analysis. As a result of the comparison through the weighted F1-score, the method using federated learning suggested the applicability of federated learning to logarithmic anomaly detection when compared to the existing learning method. Hybrid models perform better than single models for conventional learning methods and federated learning in log data anomaly detection.

However, although federated learning ensures data anonymity, it is still in the early stages of research, and more in-depth research is needed regarding security. In particular, we are exposed to new risks due to the model's parameter sharing and communication for learning, allowing the parameters and outputs of the learning model to be manipulated [24]. In the real-world environment of federated learning, models are updated through the devices of various clients, so it is necessary to select trusted clients. In consideration of these situations, future federated learning requires further studies for improvement, such as introducing indicators [25] that measure client reliability.

**Author Contributions:** Conceptualization, T.-H.S.; Methodology, T.-H.S.; Software, T.-H.S.; Validation, T.-H.S.; Formal analysis, T.-H.S.; Investigation, T.-H.S.; Resources, T.-H.S.; Data curation, T.-H.S.; Writing—original draft, T.-H.S.; Writing—review & editing, T.-H.S.; Visualization, T.-H.S.; Supervision, S.-H.K.; Project administration, S.-H.K.; Funding acquisition, S.-H.K. All authors have read and agreed to the published version of the manuscript.

**Funding:** This work was supported by an Institute for Information & Communications Technology Planning & Evaluation (IITP) grant funded by the Korean government (MSIT) (No.2022-0-01203, Regional strategic Industry convergence security core talent training business).

**Institutional Review Board Statement:** Not applicable.

**Informed Consent Statement:** Not applicable.

**Data Availability Statement:** The data presented in this study are openly available at refs [12,22].

**Conflicts of Interest:** The authors declare no conflict of interest.

## Abbreviations

The following abbreviations are used in this manuscript:

| | |
|---|---|
| FedSGD | Federated Stochastic Gradient Descent |
| FedAVG | Federated Averaging |
| TF-IDF | Term Frequency-Inverse Document Frequency |
| LSTM | Long Short Term Memory Network |
| CNN | convolutional neural network |
| HDFS | Hadoop Distributed File System |
| BGL | Blue Gene/L |
| LLNL | Lawrence Livermore National Labs |

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
