# Peer review of "Utility Analysis about Log Data Anomaly Detection Based on Federated Learning"

_applsci, doi:10.3390/app13074495_

Round 1

Reviewer 1 Report

This paper proposes a novel deep learning model for log anomaly detection based on federated learning. Authors compared federated learning to  the centralized learning method.

The main contributions are the development of federated learning based models and the development of hybrid models for log anomaly detection.

This topic is interesting and suitable for publication. 

The paper is clearly written and well organised. Experimental results are convincing. Overall, it is a good work in my view and worth publishing as a paper. I have a bit of comments.

My detailed comments are as follows:

  1. In the third contribution the two models used for hybrid models should be included.
  2. The public datasets used for the experiments should be cited.
  3. The number of weights used on each of the compared models should be included.

Author Response

Dear Reviewer,

Thank you for your comment.

Thankyou

Reviewer 2 Report

 -Paper is well organized.

-Introduction is written well.

-Successfully executed the proposed methodology.

-Results obtained are good.

-Technical quality of the paper is good.

-Figures are clear.

-References are also appropriate.

Author Response

Dear Reviewer,

Thank you for your comment.

Best regards

Author Response

(The authors gave the same response as above.)

Round 2

Reviewer 3 Report

The authors have answered all my questions and make the corresponding revisions.